# Doxifluridine promotes host longevity through bacterial metabolism

Rui Wei[1], Yuling Peng[1], Yamei Luo[1], Xinyuan Wang[2], Zhenzhong Pan[1], Ran Zhou[1,3], Huan Yang[1], Zongyao Huang[1], Yaojia Liu[1], Lunzhi Dai[1], Yuan Wang[1,3]*, Yan Zhang[1]*

1 State Key Laboratory of Biotherapy, National Clinical Research Center for Geriatrics, West China Hospital, Sichuan University, Chengdu, Sichuan, China, 2 Proteomics-Metabolomics Platform of Core Facilities, West China Hospital, Sichuan University, Chengdu, Sichuan, China, 3 Tianfu Jincheng Laboratory, Frontiers Medical Center, Chengdu, Sichuan, China

☙ These authors contributed equally to this work.
* yanzhang@scu.edu.cn (YZ); wangyuan@scu.edu.cn (YW)

**Data availability statement:** The changes in alternative splicing in nematodes induced by

## Abstract

Aging is associated with alternative splicing (AS) defects that have broad implications on aging-associated disorders. However, which drug(s) can rescue age-related AS defects and extend lifespan has not been systematically explored. We performed large-scale compound screening in *C. elegans* using a dual-fluorescent splicing reporter system. Among the top hits, doxifluridine, a fluoropyrimidine derivative, rescues age-associated AS defects and extends lifespan. Combining bacterial DNA sequencing, proteomics, metabolomics and the three-way screen system, we further revealed that bacterial ribonucleotide metabolism plays an essential role in doxifluridine conversion and efficacy. Furthermore, doxifluridine increases production of bacterial metabolites, such as linoleic acid and agmatine, to prolong host lifespan. Together, our results identify doxifluridine as a potent lead compound for rescuing aging-associated AS defects and extending lifespan, and elucidate drug's functions through complex interplay among drug, bacteria and host.

## Introduction

Disrupted transcriptional and protein homeostasis contributes to aging, characterized by progressive functional decline leading to increased mortality [1–5]. A key intermediary link between gene expression and the proteome is RNA processing, especially alternative splicing (AS) of pre-mRNA [5]. AS is a posttranscriptional process that generates multiple mRNA isoforms from a single gene, contributing to the complexity and diversity of organisms [6]. Precise regulation of AS is crucial for maintaining whole-body homeostasis and function [7,8].

It has been shown that aging is associated with alterations in AS patterns, which affects important biological processes and plays a key role in modulating aging and age-associated diseases [9–11]. In a 2017 study [9], Heintz et al. used a GFP/mCherry dual-fluorescent reporter system for the AS of *ret-1* exon 5 in *C. elegans* [12], in which aging-associated AS defects could be signaled by reduced GFP-to-mCherry ratio as the worms age. These defects could be alleviated by dietary restriction, a well-known lifespan-extending intervention [13]. Mechanistically, splicing factor 1 (SFA-1) is required for lifespan extension by dietary restriction. These data imply an exciting therapeutic opportunity that targeting AS dysregulation

aging and doxifluridine treatment raw sequence reads have been deposited in NCBI BioProject (https://www.ncbi.nlm.nih.gov/bioproject/) under accession number PRJNA1221008. Genome sequencing raw data of different E. coli strains have been deposited in the Sequence Read Archive (https://www.ncbi.nlm.nih.gov/sra/) under accession number PRJNA1070887. The mass spectrometry proteomics data have been deposited in the ProteomeXchange Consortium via the iProX partner repository (https://www.iprox.cn/page/home.html) with dataset identifier PXD049160. Metabolomics data have been deposited in Figshare repository (https://doi.org/10.6084/m9.figshare.25111361).

**Funding:** This work was supported by National Natural Science Foundation of China (82173179, 32471214 to YZ), Sichuan Science and Technology Program (2023ZYD0128, 2024NSFSC0059 to YW). The funders had no role in study design, data collection and analysis, decision to publish, or preparation of the manuscript.

**Competing interests:** The authors have declared that no competing interests exist.

may delay aging and prevent or treat age-related diseases. However, which drug(s) can rescue age-related AS defects and extend lifespan has not been systematically explored.

Microbes play a significant role in regulating host health [14–16], and the availability and efficacy of drugs can also be influenced by the gut microbiota [17–20]. Due to the complexity of interactions between bacteria and drugs within the host, the effects of drugs on host longevity need to be precisely investigated. The nematode *C. elegans*, primarily colonized by enterobacteria, serves as an excellent model for studying microbe-host interactions [21–23]. Recently, the development of three-way/four-way system (Drug-Microbe-Host/Drug-microbe-nutrient-Host) has facilitated the identification of interactions between drugs, bacteria, nutrients, and the host [24,25].

Here, we built upon the Heintz et al. study and performed large-scale compound screening in *C. elegans*, using the dual-fluorescent splicing reporter system to identify drugs that can rescue age-associated AS defects and extend lifespan. We discovered a lead compound doxifluridine, a fluoropyrimidine derivative, that rescues age-associated AS defects and dramatically extends both lifespan and healthspan. Interestingly, the effect of doxifluridine on lifespan is dependent on bacterial metabolism. By combining bacterial DNA sequencing, proteomics, metabolomics and the three-way screen system, we further revealed the complex interactions among drug and bacteria in regulating host longevity.

## Results

### Compound screen identified small molecules that rescue age-associated alternative splicing defects in *C. elegans*

We used a GFP/mCherry dual-fluorescent reporter for age-associated alternative splicing (AS) in *C. elegans*. The reporter strain, KH2235, expresses a pair of *ret-1* exon 5 reporter minigenes with differential frameshifts. GFP expression indicates inclusion of exon 5, while mCherry expression indicates exon 5 skipping [9](S1A Fig). This *ret-1* splicing event changes with aging and are regulated by splicing factor 1 (SFA-1), which is required for lifespan extension. We observed a 75% reduction of the GFP/mCherry ratio in the worm intestine at Day 7 (D7) compared to Day 1 (D1) (S1B Fig). To identify small molecules that could rescue age-associated AS defects, we screened 5,623 compounds (4,131 bioactive compounds from Tocris and 1,492 FDA-approved drugs) to find positive hits that increase GFP/mCherry ratios at D7. After three rounds of screening, we obtained 26 consistent positive hits (Fig 1A and S1 Table). We further tested the rescue effects of these compounds at varying concentrations, and identified 10 hits that rescue GFP/mCherry ratios at multiple concentrations (S1C Fig). These 10 compounds include 1 anticancer drug, 2 anti-epilepsy drugs, 1 ion channel inhibitor, 2 cosmetics, 1 immunosuppressant, and 3 bioactive compounds without known targets (S1 Table). We aim to determine whether the drug not only rescues age-associated alternative splicing (AS) defects but also extends lifespan, as the AS changes induced by the drug may not necessarily be linked to lifespan extension. We examined the effects of these AS-rescuing compounds on the lifespan of *C. elegans*, and found 3 hits that significantly prolong the lifespan (S1D–S1H Fig and S1 Table).

Among these three compounds, doxifluridine is most effective, which is a fluoropyrimidine derivative of 5-fluorouracil (5-FU) and an oral prodrug that can be metabolized into 5-FU in vivo [26]. Doxifluridine increases the GFP/mCherry ratio at D7 (Fig 1B and 1C). Another fluoropyrimidine derivative, 5-fluorodeoxyuridine (FUDR), which is commonly used to inhibit fertility in *C. elegans* [27,28], also rescues the GFP/mCherry ratio at D7 (S1I and S1J Fig). However, doxifluridine is less toxic than FUDR in normal human hepatic cell line LO2 (S1K Fig). AMPK/mTORC1 signaling acts as upstream regulators of AS homeostasis [9]. We found

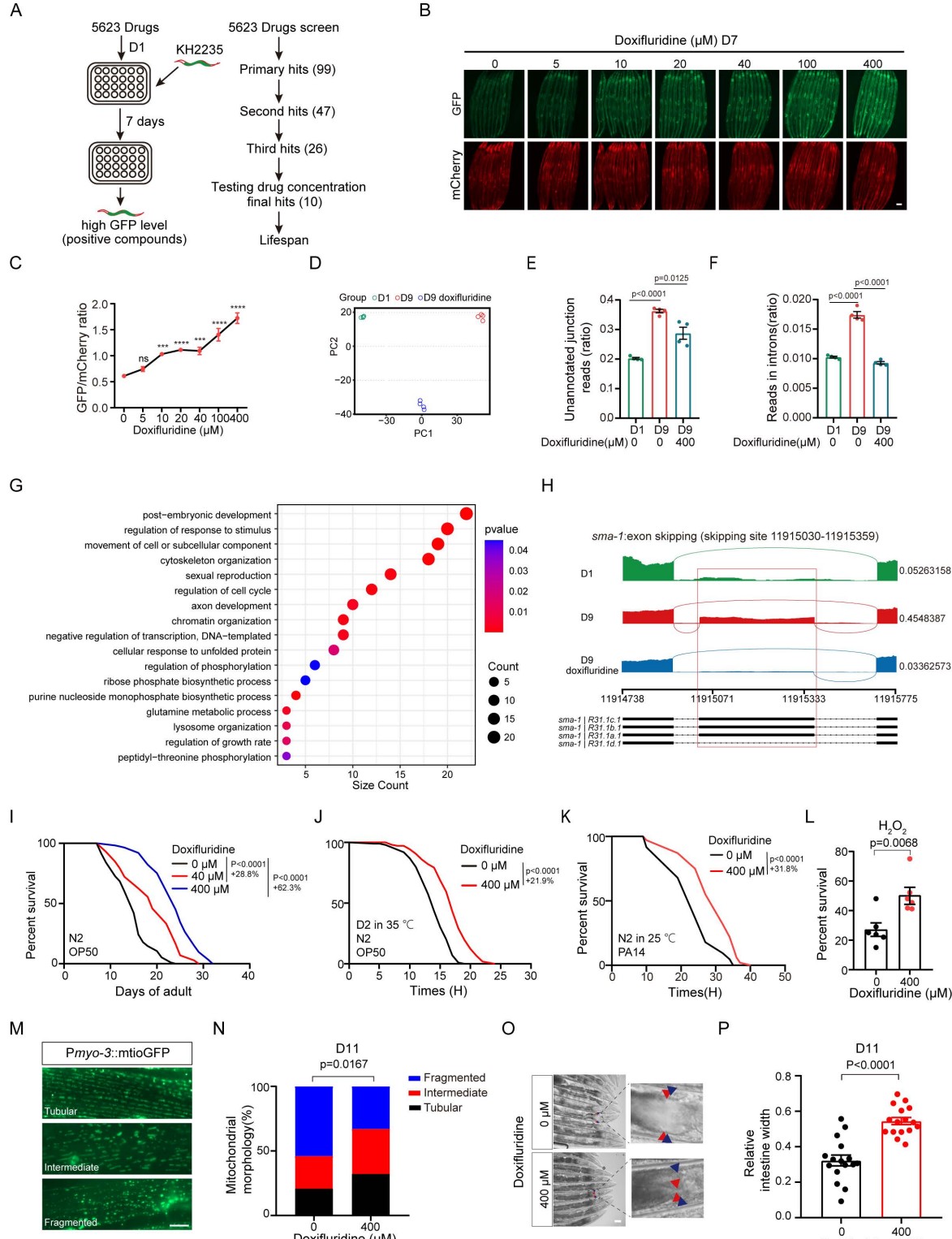

**Fig 1. Large-scale drug screen identifies doxifluridine that rescues age-associated AS defects and extends lifespan and healthspan in *C. elegans*.** (A) Diagram of 5623 compounds screen. Compounds preventing the age-associated AS defects by measuring the GFP to mCherry ratio were selected for further round testing. (B) Fluorescent images of worms with splicing reporter treated with different concentrations of doxifluridine. (C) GFP/mCherry fluorescence intensity ratio in B. 9-14 worms were imaged at D7. Each worm was measured by randomly selecting 8 regions in whole intestine (Methods). n=2 independent experiments. (D) PCA plot of splicing

genes in worms at D1, D9, and D9 with doxifluridine treatment. (E-F) Unannotated junction reads (E) and intron reads (F) in D1, D9, and D9 animals treated with 400 μM doxifluridine. (G) Gene Ontology (GO) analysis of genes with age-related alternative splicing changes reversible by doxifluridine. (H) Sequencing reads tracks for *sma-1* pre-mRNA in wild-type worms at D1, D9, and D9 with doxifluridine treatment. The numbers on the right indicate the proportion of exon (11915030-11915359) inclusion (PSI). PSI=(J1+J2)/(J1+J2+2*J3). J1 indicates reads of splice products that include the exon (exon inclusion); J2 refers to reads of another splice product that includes the exon (may involve other splicing events); J3 represents reads of splice products that skip the exon (exon skipping). (I) Survival analysis of doxifluridine-treated and vehicle-treated worms. n=2 independent experiments. (J-L) Stress assays of *C. elegans* treated and non-treated with doxifluridine. Resistance to heat stress (J), pathogenic challenge to *Pseudomonas aeruginosa* (PA14) (K) and resistance to oxidative stress (L). n=2 independent experiments. (M) Representative fluorescent images of three types of mitochondria morphology labeled by mitoGFP in the body wall muscles. n=2 independent experiments. (N) Distribution of different types of mitochondria morphology in doxifluridine-treated and vehicle-treated worms. 11~15 D11 worms were imaged and approximately 130 muscle cells were assayed per group. n=2 independent experiments. (O) Intestinal images on D11 of *C. elegans* treated with or without doxifluridine 400 μM. n=2 independent experiments. (P) The relative intestinal width statistic of O. n=2 independent experiments. The lifespan data illustrated in the figure corresponds to one repeat experiment. Error bars, SEM. *$P < 0.05$, **$P < 0.01$, ***$P < 0.001$, ****$P < 0.0001$. C, one-way ANOVA. E, F, L, P, unpaired two-tailed Student's t test. I, J, K, Log-rank (Mantel-Cox) test. N, Chi-square test. B, scale bars,100 μm; M, scale bars,10 μm; O, scale bars, 50 μm.

that metformin and rapamycin which are AMPK activating drugs and mTORC1-inhibiting drugs could rescue age-associated AS defects, while doxifluridine is more potent than metformin and rapamycin in reversing the age-associated AS defects (S1L Fig).

To investigate changes in alternative splicing events during aging and in response to drug administration, we performed RNA-seq experiments on young worms (D1), aged worms (D9), aged worms treated with doxifluridine. We found that aging led to the dysregulated alternative splicing, characterized by elevated intron retention and unannotated splice junctions on D9 compared to D1, while doxifluridine treatment significantly reversed these alterations, restoring splicing patterns to levels comparable to D1 (Fig 1D–1F and S2 Table). Additionally, we conducted Gene Ontology (GO) analysis on genes showing aging-related changes in alternative splicing that could be reversed by doxifluridine treatment. The enriched pathways highlighted several critical cellular functions, including cytoskeleton organization, regulation of the cell cycle, chromatin organization, and more (Fig 1G and S2 Table). For instance, aging leads to an increase in exon inclusion rates of *sma-1*, a gene involved in cell shape formation [29], which can be reversed by treatment with doxifluridine (Fig 1H). These data suggest that doxifluridine mitigates age-related dysfunctional processing of specific pre-mRNAs.

We observed that doxifluridine treatment significantly extends the lifespan of nematodes (Fig 1I). We also found that doxifluridine protects *C. elegans* from heat stress, pathogen (PA14) infection, and oxidative stress (Fig 1J–1L and S3 Table). In addition, muscle mitochondria and intestine show dramatic alterations at aged days (D11) [30–32]. We found that doxifluridine ameliorated muscle mitochondrial fragmentation and intestinal atrophy in aged worms (Fig 1M–1P). These results indicate that doxifluridine improves both lifespan and healthspan in *C. elegans*.

## Lifespan extension by doxifluridine requires live bacteria

Previous studies have reported that bacterial metabolism affects the *C. elegans* response to fluoropyrimidines [28,33], so we investigated whether lifespan extension by doxifluridine requires live bacteria. We treated worms fed with freeze-dried bacteria powder or live bacteria using both low (40 μM) and high (400 μM) drug concentrations. At both concentrations, the lifespan extension by doxifluridine is dramatically reduced in the dead bacteria group (Groups with live OP50 (doxifluridine 40 μM vs 0 μM) showed a 31% increase (p < 0.0001) and (doxifluridine 400 μM vs 0 μM) a 49.4% increase (p < 0.0001). Groups with freeze-dried

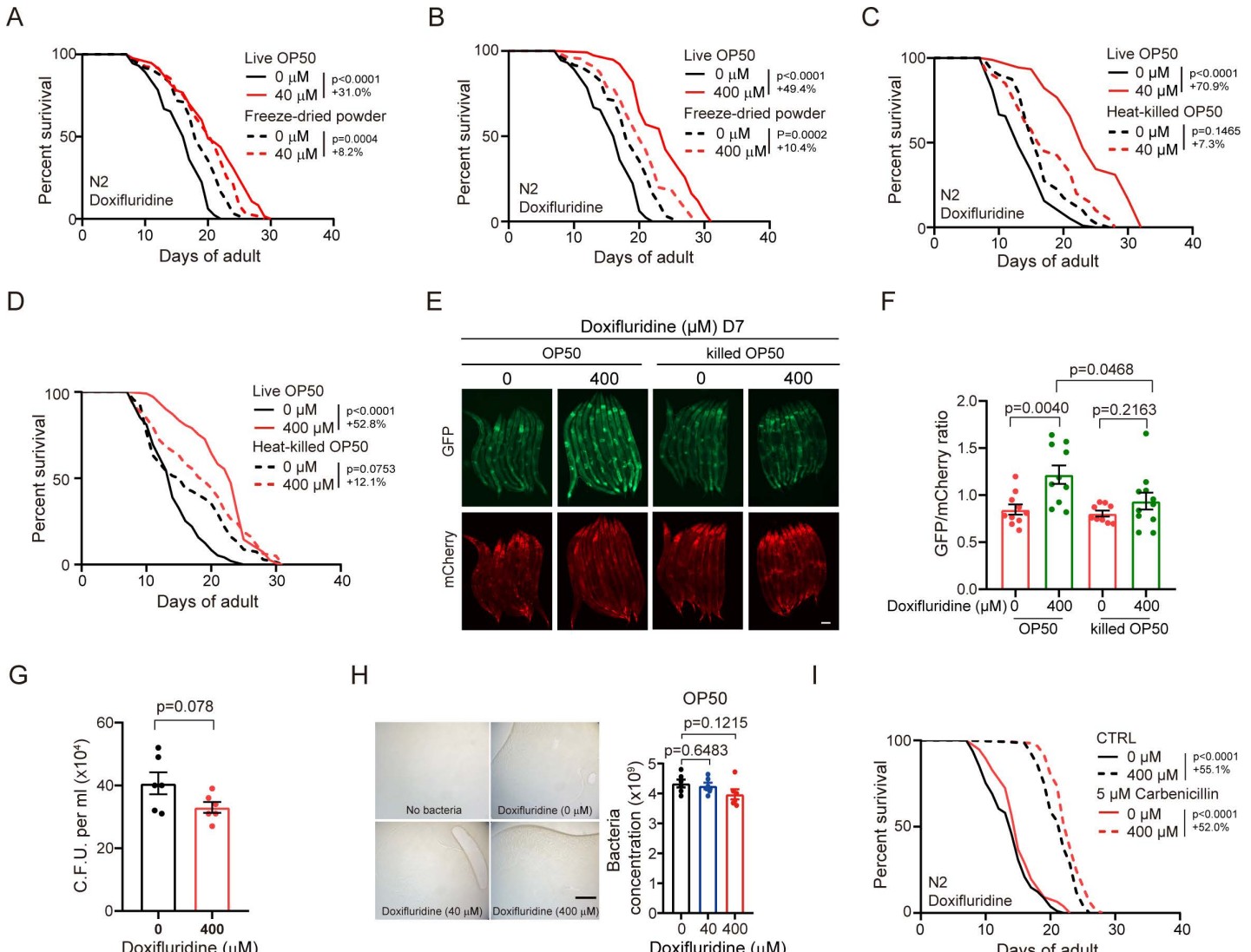

**Fig 2. Lifespan extension by doxifluridine requires live bacteria.** (A and B) Lifespan curves of worms treated with or without doxifluridine 40 μM (A), 400 μM (B) in freeze-dried powder bacteria and live bacteria. Lifespan curves of live OP50 and freeze-dried powder without doxifluridine treatment were shown twice for comparison in (A) and (B). n=2 independent experiments. (C and D) Lifespan curves of worms treated with or without doxifluridine 40 μM (C), 400 μM (D) in heat-killed OP50 and live OP50. n=2 independent experiments. (E) Fluorescent images of worms with splicing reporter treated with or without doxifluridine 400 μM in OP50 and heated-killed OP50. (F) GFP/mCherry fluorescence intensity ratio in (E). 9-11 worms were imaged at D7. n=2 independent experiments. (G) The colony number of subcultured bacteria treated with or without doxifluridine. Combination of two independent experiments. n=2 independent experiments. (H) Left: representative images of bacterial lawn morphology on a solid petri dish treated with or without doxifluridine. Right: concentration of bacteria with or without doxifluridine treatment. Combination of two independent experiments. n=2 independent experiments. (I) Lifespan curves of worms treated with or without doxifluridine 400 μM in OP50 and OP50 with 5 μM carbenicillin. n=2 independent experiments. The lifespan data illustrated in the figure corresponds to one repeat experiment. Error bars, SEM. A-D, I, Log-rank (Mantel-Cox) test. F, G, H, unpaired two-tailed Student's t test. E, scale bars,100 μm. H, scale bars, 0.5 cm.

OP50 (doxifluridine 40 μM vs 0 μM) showed an 8.2% increase (p = 0.0004) and (doxifluridine 400 μM vs 0 μM) a 10.4% increase (p = 0.002)) (Fig 2A and 2B). Similarly, heat- or UV-killed bacteria also greatly reduce the effect of doxifluridine on lifespan extension (Groups with live OP50 (doxifluridine 40 μM vs 0 μM) showed a 70.9% increase (p < 0.0001) and (doxifluridine 400 μM vs 0 μM) a 52.8% increase (p < 0.0001). Groups with heat-killed OP50 (doxifluridine 40 μM vs 0 μM) showed a 7.3% increase (p = 0.1465) and (doxifluridine 400 μM vs 0 μM) a

12.1% increase (p = 0.0753)) (Figs 2C and 2D and S2A). Pretreating *E. coli* with doxifluridine also increases lifespan (S2B Fig). These results suggest that doxifluridine requires live bacteria to extend the lifespan of *C. elegans*. We then examined whether the AS-related splicing reporter changes in KH2235 induced by doxifluridine depend on the presence of live bacteria. Our results showed that the higher ratio of GFP/mCherry induced by doxifluridine was completely blocked in heat-inactivated bacteria group, suggesting that live bacteria are required for the alternative splicing changes by doxifluridine treatment (Fig 2E and 2F).

Feeding on nonproliferating bacteria can extend the lifespan of nematodes [34]. At a high concentration (400 μM), doxifluridine exhibited moderate inhibition of bacterial proliferation (S2C Fig). We tested whether the effect of doxifluridine on bacteria is bactericidal or bacteriostatic. We subcultured the bacteria from doxifluridine-treated plates and found that *E. coli* shows no significant reduction in colony-forming units (Figs 2G and S2D). Additionally, the morphology and quantity of bacteria treated with doxifluridine were comparable to those of untreated bacteria (Fig 2H). These results demonstrate that doxifluridine does not exhibit bactericidal activity. We then investigated whether the lifespan-extending effect of the doxifluridine depend on its antibiotic effect. To test this, we assessed the drug's effect in the presence of carbenicillin, a classic antibiotic that inhibits bacterial proliferation [35]. While carbenicillin suppressed bacterial growth more effectively than doxifluridine (S2C Fig), it resulted in less lifespan extension compared to doxifluridine (Fig 2I). Importantly, doxifluridine still extended worm lifespan even in the presence of carbenicillin-induced bacterial inhibition (Fig 2I), indicating that its lifespan-extending effect does not rely on its antibacterial activity.

## Lifespan extension by doxifluridine requires bacterial ribonucleotide metabolism

Various bacterial strains differentially affect the *C. elegans* response to therapeutics [24,28,33,36]. To test this possibility, we fed worms with six commonly used *E. coli* strains including OP50, BL21, MG1655, BW25113, X1666, HT115, and compared the efficacy of doxifluridine treatment. We found that doxifluridine treatment achieves greatest lifespan extension in the OP50 group which is the most significant at the lower concentration, while showing a trend at the higher concentration (Figs 3A and S2E). To identify strain-specific genetic factors that influence drug efficacy, we performed whole genome sequencing of these six bacterial strains, and predicted their evolutionary relationship based on single nucleotide polymorphism (SNPs). We found that OP50 genetically differs from the other five strains, harboring unique mutations in 206 genes which may influence the efficacy of doxifluridine (Fig 3B and S4 Table).

The downstream metabolite of doxifluridine, 5-FU, requires the pyrimidine metabolic pathway to exert its function [37,38]. Indeed, among the 206 genes uniquely mutated in OP50, we found one gene in the salvage pathway of pyrimidine synthesis, *yjjG* (Fig 3C). *yjjG* is a pyrimidine 5'-nucleotidase that converts FUMP to FUrD [25] (Fig 3D). SNPs in *yjjG* results in an early stop codon mutation. To test the functional impact of *yjjG*-loss, we overexpressed wildtype *yjjG* in OP50. The lifespan extension effect by doxifluridine is partially diminished in worms fed with *yjjG*-overexpressing OP50 (Groups with live OP50 (doxifluridine 400 μM vs 0 μM) showed a 40.6% increase (p < 0.0001), while *yjjG*-overexpressing OP50 had a 14.6% increase (p < 0.0001)) (Fig 3E).

Mechanistically, *yjjG* mutation in OP50 likely leads to increased level of FUMP. To identify the bacterial metabolic enzymes involved in the efficacy of doxifluridine, we utilized a commercial library of BW25113 mutant strains. Single knockout (KO) of *upp* or *udp* in the BW25113 strain (Keio collection library), which are required for 5-FU to FUMP conversion (Fig 3D), partially reduces the efficacy of doxifluridine on lifespan, which could be restored by

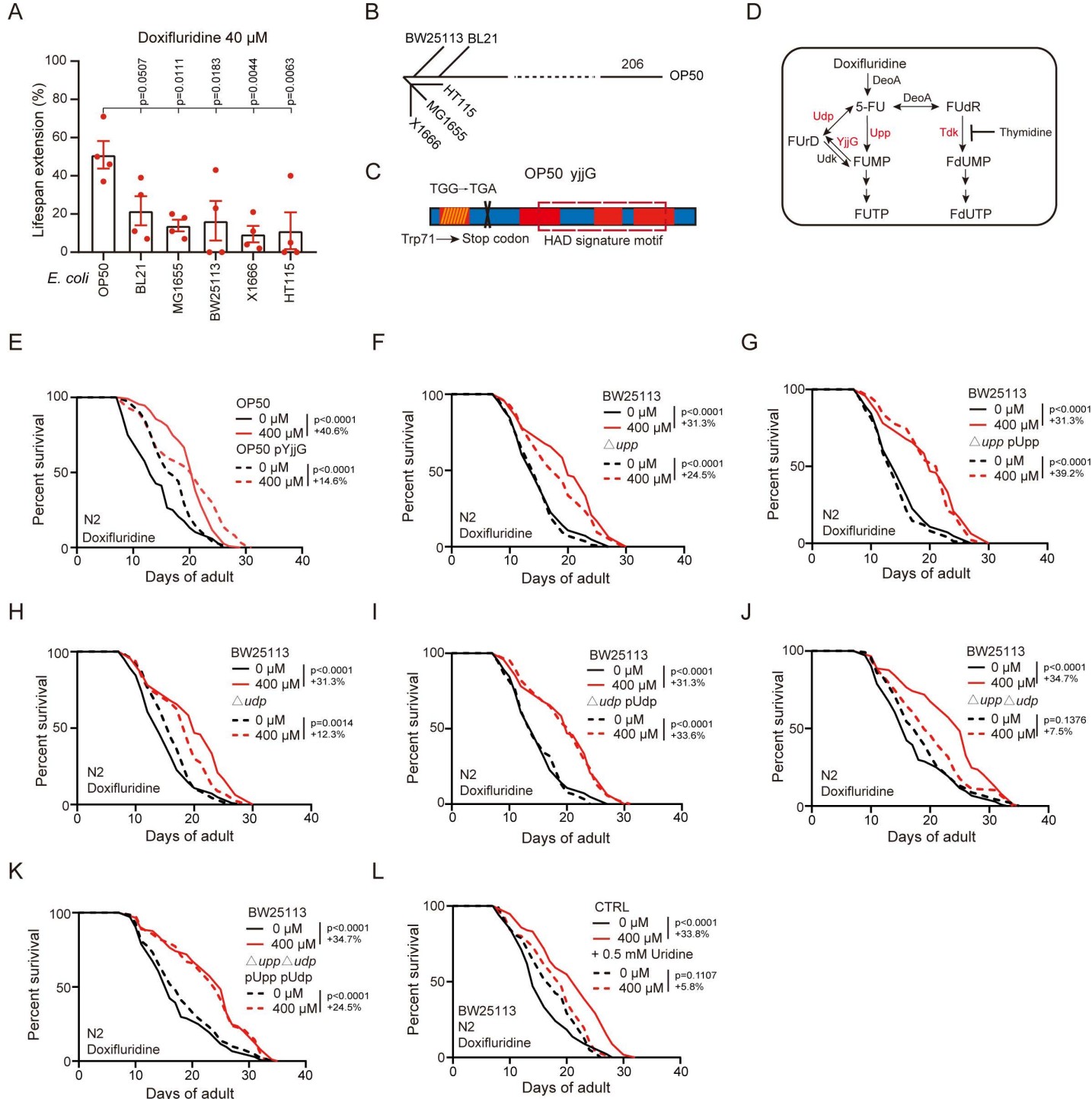

**Fig 3. Lifespan extension by doxifluridine requires bacterial ribonucleotide metabolism.** (A) Mean lifespan extension of worms treated with doxifluridine (40 μM) in different laboratory bacteria strains. *E. coli* K-12 strains: BW25113, HT115, MG1655, X1666; *E. coli* B strains: OP50, BL21. Summary of four independent experiments, with each point representing one replicate in Fig 3A. (B) Predicted evolutionary relationship among *E. coli* strains based on SNPs, using public MG1655 genome as a reference. (C) Schematic representation of *yjjG* gene mutation sites in OP50. (D) Diagram of fluoropyrimidine metabolism in *Escherichia coli*. (E) Lifespan curves of worms treated with or without doxifluridine 400 μM in OP50 and OP50 overexpressing YjjG. n=2 independent experiments. (F) Lifespan curves of worms treated with or without doxifluridine 400 μM in BW25113 and BW25113 △*upp*. n=2 independent experiments. (G) Lifespan curves of worms treated with or without doxifluridine 400 μM in BW25113 and BW25113 △*upp* with overexpressing Upp. n=2 independent experiments. (H) Lifespan curves of worms treated with or without doxifluridine 400 μM in BW25113 and BW25113 △*udp*. n=2 independent experiments. (I) Lifespan curves of worms treated with or without doxifluridine 400 μM in BW25113 and BW25113 △*udp* with overexpressing Udp. n=2 independent experiments. Lifespan curves of control group (BW25113) treated with

and without doxifluridine were the same in (F), (G), (H), (I). (J) Lifespan curves of worms treated with or without doxifluridine 400 μM in BW25113 and BW25113 Δ*upp*Δ*udp*. n=2 independent experiments. (K) Lifespan curves of worms treated with or without doxifluridine 400 μM in BW25113 and BW25113 Δ*upp*Δ*udp* with overexpressing Upp and Udp. n=2 independent experiments. Lifespan curves of BW25113 treated with and without doxifluridine were shown twice in (J) and (K) for comparison. (L) Lifespan curves of worms treated with or without doxifluridine 400 μM in BW25113 and BW25113 with uridine supplement. n=2 independent experiments. The lifespan data illustrated in the figure corresponds to one repeat experiment. Error bars, SEM. A, one-way ANOVA. E-L, Log-rank (Mantel-Cox) test.

overexpression *upp* or *udp* (Groups with BW25113 (doxifluridine 400 μM vs 0 μM) showed a 31.3% increase (p < 0.0001), Δ*upp* (24.5%, p < 0.0001), Δ*upp* pUpp (39.2%, p < 0.0001), Δ*udp* (12.3%, p =0.0014), and Δ*udp* pUdp (33.6%, p < 0.0001)) (Fig 3F–3I). Double KO of *upp* and *udp* completely reduces the efficacy of doxifluridine, which could also be rescued by overexpression of *upp* and *udp* (Groups with BW25113 (doxifluridine 400 μM vs 0 μM) showed a 34.7% increase (p < 0.0001), Δ*upp*Δ*udp* (7.5%, p = 0.1376), Δ*upp*Δ*udp* Pupp pUdp (24.5%, p < 0.0001)) (Fig 3J and 3K). In contrast, KO of *tdk*, a nucleoside diphosphate kinase from the deoxyribonucleotide salvage pathway [25,33] (Fig 3D), did not influence the efficacy of doxifluridine (S2F Fig). Furthermore, the increased GFP/mCherry ratio is markedly reduced in Δ*upp*Δ*udp* compared to wildtype BW25113 (The relative GFP/mCherry ratio increased by 36.4% in the BW25113 (doxifluridine 400 μM vs 0 μM) group, and by 24.2% in the Δ*upp*Δ*udp* (doxifluridine 400 μM vs 0 μM) group, a 12.2% decrease in Δ*upp*Δ*udp* compared to BW25113) (S2G and S2H Fig).

To further verify the effect of bacterial ribonucleotide metabolism on the efficacy of doxifluridine on lifespan, we added five types of nucleobases separately onto the *E. coli* plates (S2I Fig). We found that uridine supplementation completely abolishes the lifespan extension by doxifluridine (Groups with BW25113 (doxifluridine 400 μM vs 0 μM) showed a 33.8% increase (p < 0.0001), while supplementation of 0.5 mM uridine showed a 5.8% increase (p = 0.1107)) (Figs 3L and S2J), while thymidine supplementation further increases the lifespan (S2K Fig). Supplementation of the other three types of nucleobases does not have a significant effect. Together, these data support that ribonucleotide metabolism but not deoxyribonucleotide metabolism is necessary for doxifluridine-mediated lifespan extension.

## Bacterial metabolites induced by doxifluridine contribute to host longevity

It is also possible that doxifluridine treatment could induce bacteria to produce metabolites that extend the host lifespan. To test this, we performed bacterial proteomics and metabolomics to identify metabolic changes upon doxifluridine treatment. We found that doxifluridine treatment upregulates TCA cycle, nitrogen metabolic pathways, while downregulating glycolysis, arginine and proline metabolic pathway (Fig 4A–4C and S5 Table).

These altered metabolic pathways are regulated by key upstream transcription factors predicted by RegulonDB database [39] (S3A Fig and S6 Table, Methods). To pinpoint which of the metabolic pathways are essential for the efficacy of doxifluridine, we compared the treatment effect using bacteria harboring KO mutations in these transcription factors. We also used the BW25113 mutant strain in Keio collection library. We found that KO of two transcription factors regulating nitrogen metabolism (*adiY*, *lrp*) completely reduce the drug efficacy (Groups with BW25113 (doxifluridine 400 μM vs 0 μM) showed a 30.1% increase (p < 0.0001), Δ*adiY* (-6%, p = 0.8243), Δ*adiY* pAdiY (39.9%, p < 0.0001); Groups with BW25113 (doxifluridine 400 μM vs 0 μM) showed a 37.8% increase (p < 0.0001), Δ*lrp* (-6.4%, p = 0.3083), Δ*lrp* pLrp (26.2%, p < 0.0001)) (Figs 4D–4G and S3B). The protein levels of glutamate synthase subunits *gltB* and *gltD,* which are direct downstream targets of AdiY and LRP and positive regulators of nitrogen metabolism, is increased in doxifluridine-treated bacteria

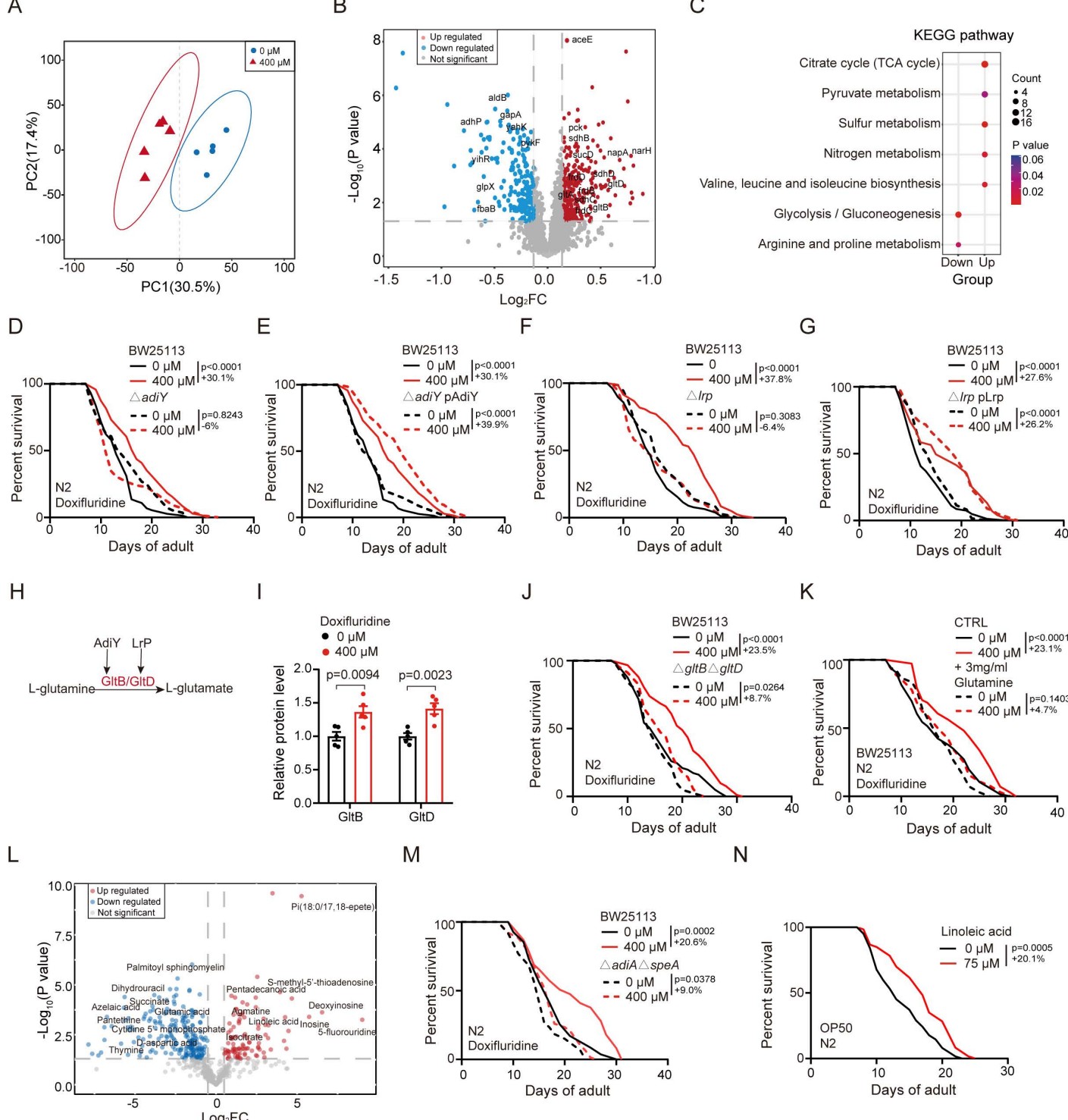

**Fig 4. Bacterial metabolites induced by doxifluridine contribute to host longevity.** (A) PCA plot of *E. coli* OP50 proteomics data showing effects of doxifluridine treatment or non-treatment in *E. coli* OP50. (B) Volcano plots of *E. coli* OP50 proteomics data showing distinct protein signatures associated with doxifluridine treatment. Highlighted proteins are differential expressed proteins which are analyzed in KEGG pathway. (C) KEGG pathway enrichment using differential expressed proteins in (B). (D and E) Lifespan curves of worms treated with or without doxifluridine 400 μM in BW25113, BW25113 △*adiY* (D) and BW25113 △*adiY* with overexpressing AdiY (E). n=2 independent experiments. Lifespan curves of BW25113 treated with and without doxifluridine were shown twice in (D) and (E) for comparison. (F) Lifespan curves of worms treated with or without doxifluridine 400 μM in BW25113 and BW25113 △*lrp*. n=2 independent experiments. (G) Lifespan curves of worms treated with or without doxifluridine 400 μM in BW25113 and BW25113 △*lrp* with overexpressing Lrp. n=2 independent

experiments. (H) Diagram of transcription factor AdiY/Lrp regulating the expression of GltB/GltD which is related to glutamine metabolic pathway. (I) Relative protein levels of GltB and GltD in *E. coli* OP50 treated with 400 μM doxifluridine compared with vehicle treatment. (J) Lifespan curves of worms treated with or without doxifluridine 400 μM in BW25113 and BW25113 △*gltB*△*gltD*. n=2 independent experiments. (K) Lifespan curves of worms treated with or without doxifluridine 400 μM in BW25113 with or without glutamine supplementation. n=2 independent experiments. (L) Volcano plots of *E. coli* OP50 metabolomics data showing distinct metabolic signatures associated with doxifluridine 400 μM treatment. (M) Lifespan curves of worms treated with or without doxifluridine 400 μM in BW25113 and BW25113 △*adiA*△*speA*. n=2 independent experiments. (N) Lifespan curves of worms treated with or without linoleic acid supplementation. n=2 independent experiments. The lifespan data illustrated in the figure corresponds to one repeat experiment. Error bars, SEM. D-G, J-K, M-N, Log-rank (Mantel-Cox) test. I, unpaired two-tailed Student's t test.

(Fig 4H and 4I and S6 Table). Double KO of *gltB* and *gltD* also dramatically reduces the drug efficacy similarly to *adiY* or *lrp* KO (Groups with BW25113 (doxifluridine 400 μM vs 0 μM) showed a 23.5% increase ($p < 0.0001$), Δ*gltB*Δ*gltD* showed an 8.7% increase ($p = 0.0264$)) (Fig 4J). Since KO of *gltB* and *gltD* cause accumulation of glutamine [40], we tested whether adding excessive glutamine could reduce the drug efficacy. Indeed, glutamine addition completely blocks the effect of doxifluridine on lifespan (Groups with BW25113 (doxifluridine 400 μM vs 0 μM) showed a 23.1% increase ($p < 0.0001$), while supplementation of 3 mg/ml glutamine showed a 4.7% increase ($p = 0.1403$)) (Figs 4K and S3C). Similarly, the increased GFP/mCherry ratio observed in the doxifluridine-treated group was significantly suppressed in Δ*gltB*Δ*gltD* compared to wildtype BW25113, as well as by glutamine supplementation compared to the vehicle control (S2G, S2H, S3F and S3G Figs). Together, these data suggest that doxifluridine-induced alterations in bacterial nitrogen metabolism are necessary for its efficacy.

Furthermore, metabolomics reveals alterations in bacterial metabolites upon doxifluridine treatment (Figs 4L and S3D). As a positive control, the doxifluridine metabolite 5-fluorouridine increased 520-fold in doxifluridine-treated bacteria (Fig 4L and S7 Table). Agmatine, which has previously been shown as a bacterial metabolite to extend lifespan [24], is also among the top increased metabolites (Fig 4L and S7 Table). This is consistent with the proteomics data that arginine-agmatine degradation pathway is reduced in doxifluridine-treated bacteria (Fig 4B). Double KO of agmatine synthesis enzyme *adiA* and *speA* [41,42] completely blocks the efficacy of doxifluridine (Groups with BW25113 (doxifluridine 400 μM vs 0 μM) showed a 20.6% increase ($p = 0.0002$), while Δ*adiA*Δ*speA* showed a 9% increase ($p = 0.0378$)) (Fig 4M), suggesting that lifespan-extension effect of doxifluridine requires bacteria-derived agmatine. To investigate whether other metabolites could also play a role, we tested the lifespan-extension effects of four top metabolites (Deoxyinosine, Inosine, Methysticin and Linoleic acid) that are commercially available. Among them, only linoleic acid could significantly extend lifespan (Figs 4N and S3E). Furthermore, the addition of linoleic acid and agmatine significantly enhances the AS reporter GFP/mCherry ratio (S3H and S3I Fig). Together, these results demonstrate that doxifluridine-induced bacteria metabolites contribute to host longevity (Fig 5).

## Discussion

RNA splicing homeostasis is crucial for health and longevity, and rescuing age-associated AS defects may offer therapeutic opportunities to extend lifespan and healthspan. In this study, we identified a novel AS defect-rescuing lead compound, doxifluridine through compound screening in *C. elegans*, and elucidated that its efficacy relies on bacterial metabolism to extend lifespan.

It is conceivable that AMPK activating drugs such as metformin and mTORC1-inhibiting drugs such as rapamycin could rescue age-associated AS defects, but the AS-rescuing effect of these drugs has not been experimentally tested. To our excitement, in a side-by-side

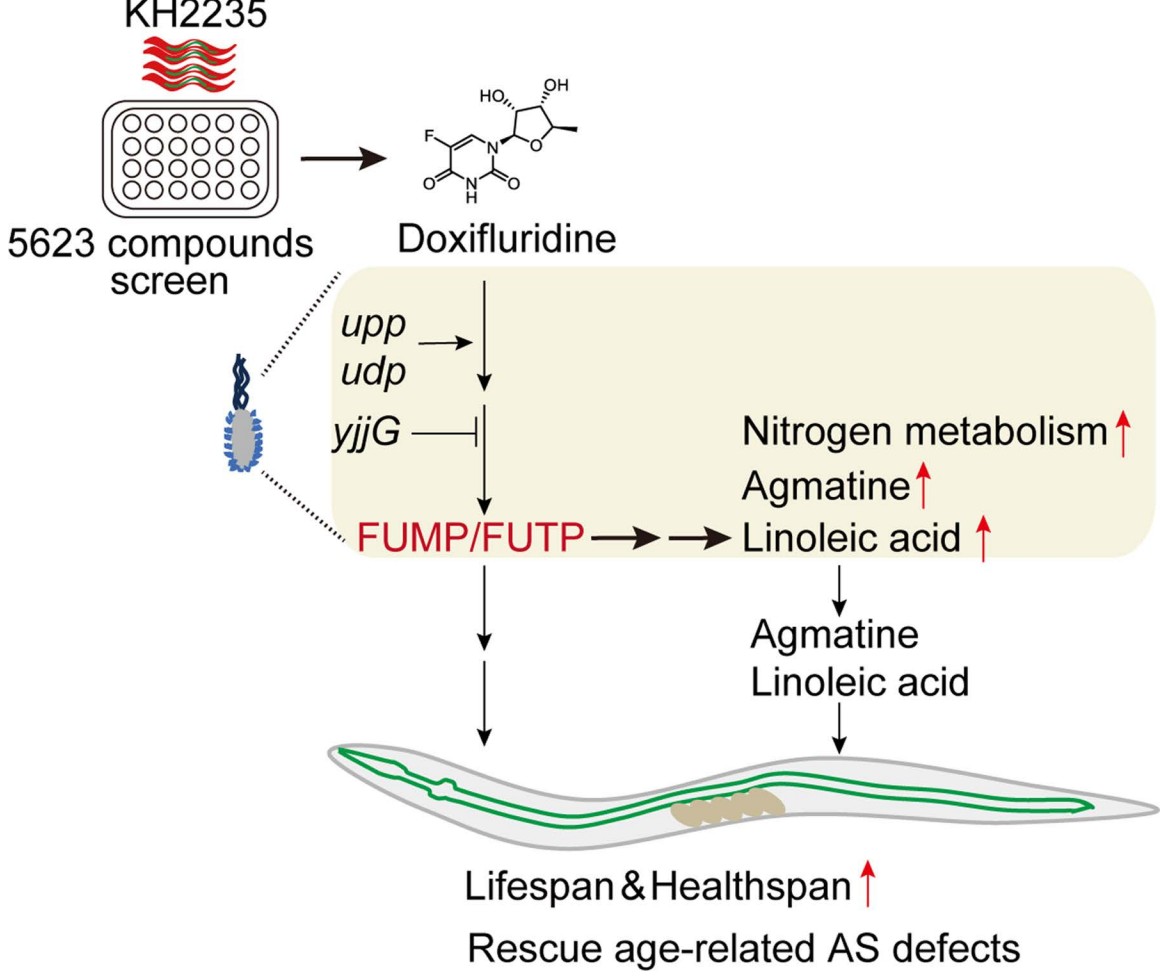

**Fig 5. Model summarizing the effects of doxifluridine on the host lifespan through bacteria metabolism.**

comparison, we found that doxifluridine is more potent than metformin and rapamycin in reversing the age-associated AS defects. Moreover, doxifluridine is less toxic and outperforms another fluoropyrimidine, FUDR, in correcting the AS defects. Notably, doxifluridine is orally available and has entered into several Phase II and Phase III clinical trials in elderly patients and demonstrated its safety [43–46]. These data support that doxifluridine is a potent lead compound to rescue age-associated AS defects and prolong survival.

The gut microbiota can modulate the availability and efficacy of drugs on host health by altering their metabolic interconversion and pharmacodynamics [17–20]. It has been previously shown that the efficacy of fluoropyrimidine drugs, such as 5-FU and FUDR, on *C. elegans* egg hatching is influenced by microbe-mediated drug metabolism. Yet little is known about the reliance of how microbes modulate the efficacy of doxifluridine, another fluoro-pyrimidine derivative, on lifespan. We used a three-way[14] screening system to investigate the interactions among doxifluridine, microbes and host lifespan in *C. elegans*. We found that doxifluridine relies on bacterial ribonucleotide metabolism to be converted to FUMP/FUTP, which are the effective drug metabolites that extend lifespan in *C. elegans*.

Conversely, drugs can also affect the metabolism of the gut microbiota to exert their efficacy. Metformin is the first lifespan-extending drug that has been shown to function through

enhanced microbial agmatine production. Similarly, doxifluridine also increases the level of bacterial agmatine, as well as other lifespan-extending bacterial metabolites such as linoleic acid. In addition, through proteomic analysis and genetic validation, we found doxifluridine-induced alterations in bacterial nitrogen metabolism are necessary for its efficacy. Thus, our results further validate the essential roles of microbiota in drug efficacy, and offer new insights into how drug-induced bacterial metabolism promotes host longevity.

In summary, our results identify doxifluridine as a potent lead compound for rescuing aging-associated AS defects and extending lifespan, and demonstrate its functions through complex interplay among drug, bacteria and host.

## Materials and methods

### C. *elegans* strains and maintenance

All *C. elegans* strains were cultured on standard nematode growth medium (NGM) plates seeded with *Escherichia coli* OP50 or BW25113 at 20 °C. All experimental consumables (pipe tips, pipettes, petri dishes, etc.) are sterile before use and regularly irradiated with UV to ensure a sterile environment. All strains were maintained contamination-free for at least two generations prior to the beginning of the experiment. Bristol N2 strain was used as the wild-type strain. KH2235(*lin-15(n765)*; *ybIs2167* [*eft-3p::ret-1E4E5(+1)E6-GGS6-mCherry+eft-3p::ret-1E4E5(+1)E6(+2)GGS6-GFP+lin-15(+)+pRG5271Neo*] X) strain was kindly gifted by William Mair (Harvard T. H. Chan School of Public Health). Other worm strains were obtained from the *Caenorhabditis* Genetic Center. All the worm strains are listed in S8 Table.

### Drug screen

5623 compounds containing 4131 compounds (TargetMol Chemicals Inc. Cat#L5600) and 1492 compounds (Selleckchem, Cat#L2000) were plated into 24-well plates with 2 mL NGM seeded with OP50 per well at a final concentration of 20 μM. Nematodes were synchronized using bleach buffer (1.6M NaOH, 12%NaClO). 20–30 synchronized young adults were plated per well and incubated at 20 °C. Worms were transferred to fresh drug screen plates daily to separate from their progeny. Images were conducted and viewed by Olympus cellSens Dimension software at D7. In the first two rounds of drug screening, we identified drugs with noticeably higher green fluorescence intensity as positive candidates. In subsequent experiments, we assessed the GFP/mCherry ratio of these positive compounds at D7 to further evaluate their efficacy. Compounds that exhibited an increased GFP/mCherry ratio were considered positive hits and validated through two additional rounds of screening. Positive hits from three rounds of screening were further tested for optimal concentration, and those from four rounds of screening were tested for lifespan.

### Microscopy and imaging analysis

For fluorescent images, worms were anesthetized with 1 mM levamisole (Sigma, L9756) and subsequently imaged using an Olympus BX63 automatic fluorescence microscope. For bright field images, an Olympus DP80 microscope with a 20× objective (UPlanSAPO) was used. Intestinal atrophy measurements were analyzed using ImageJ.

In the experiment to quantify the GFP/mCherry ratio in KH2235, images were processed and viewed using Olympus cellSens Dimension software. The images were converted to grayscale mode. The fluorescence intensity of each nematode was measured by randomly selecting eight regions (ROIs, 8×60 pixel×60 pixel) within the intestine and calculating the average grayscale value. In areas without nematodes, three random positions (3×60 pixel×60 pixel) were selected, and their mean fluorescence intensity was measured. The average value

of these measurements was used as the background fluorescence intensity for the sample. Subsequently, the average background fluorescence intensity was subtracted from the average intestinal fluorescence intensity to correct for the background, yielding the mean pixel intensity of the nematode. All experimental data associated with the GFP/mCherry ratio were replicated in two independent biological experiments.

### RNA-seq data processing and differential gene expression analysis

The raw FASTQ files were processed using Trimmomatic [47] to remove adapter sequences and low-quality reads, yielding high-quality clean data. The filtered reads were then aligned to the *C. elegans* genome (WBcel235 annotation) using STAR [48] with default parameters, generating sorted BAM files and splice junction quantification for downstream alternative splicing analysis. For differential gene expression analysis, we used DESeq2 [49] to perform normalization and differential expression test on the read counts for each gene which were estimated using Salmon [50] with the parameter --STRAND=SECOND_READ_TRAN-SCRIPTION_STRAND. Differentially expressed genes (DEGs) were identified based on a $p$ value $< 0.05$ and an absolute $\log_2$(fold-change) $> 1$. The Gene Ontology (GO) enrichment analysis was performed by clusterProfiler [51].

### Alternative splicing analysis

Genome-wide splicing efficiency was assessed by calculating the percentage of unannotated junction reads and intronic reads, following the methodology outlined in Heintz et al. [9]. Splice junctions were included in the analysis if they contained a canonical intron motif (GT/AT, CT/AC, GC/AG, CT/GC, AT/AC, GT/AT) and an overhang of at least 15 nucleotides (10% of the read length). The percentage of intronic reads was calculated using the CollectRnaSeqMetrics module from Picard [52] with the parameter STRAND=SECOND_READ_TRANSCRIPTION_STRAND. Alternative splicing (AS) events were identified and quantified using the ASpli [53] package with default parameters. The visualization of AS events was performed using Trackplot [54].

### Lifespan analysis

Lifespan assays were performed at 20°C. Nematodes were synchronized using bleach buffer (1.6M NaOH, 12%NaClO). More than 80 synchronized young adults were placed to fresh NGM plates. During the subsequent reproductive period, worms were transferred to fresh plates daily. Nematodes were considered dead when they stopped pumping and did not respond to a platinum wire touch and removed from the analysis if they crawled off the plate or displayed a bagging or protruding vulva phenotype. Each lifespan experiment included at least two independent replicates, with more than 60 animals per group for analysis. Statistical analysis was performed by the log-rank test using GraphPad Prism 8.0 software. The statistical analysis and replicate data for lifespan are in S3 Table.

To test the effect of the compound on lifespan, doxifluridine (TOPSCIENCE, Cat#T1600) was dissolved in water (15 mM stock solution) and then added to the NGM plates seeded with bacteria. Glutamine powder (Solarbio, Cat#G8230) was added into NGM media at a final concentration of 3 mg/mL in the NGM. Other additives (0.5 mM uridine (TOPSCIENCE, Cat#T2221) and 0.5 mM thymidine (TOPSCIENCE, Cat#TWP2911)) were dissolved in DMSO and then added to NGM plates seeded with OP50 bacteria. Lifespan curves of worms treated with methysticin (91.15 µM) (TOPSCIENCE, Cat#TQ0276), deoxyinosine (436 µM) (TOPSCIENCE, Cat#T1709), inosine (913.35 µM) (TOPSCIENCE, Cat#T0437), linoleic acid (75 µM) (TOPSCIENCE, Cat#T4P2931) or vehicle. The compounds were dissolved in DMSO,

with the final DMSO concentration kept at or below 0.5%, which is within the safe range for the nematodes.

## Stress assays

For heat-shock stress assay, about 120 synchronized D1 worms were added to a petri dish with or without drugs, cultured at 20 °C for 24 hours, and then transferred to 35 °C. The number of deaths was counted every hour until all worms died. This experiment was repeated twice.

For pathogen bacterial infection assay, overnight broth culture (200 µL) of the PA14 pathogenic bacterial was spread on whole surface of a 6 cm diameter PGS (fast killing) agar plate and incubated at 37 °C for 24 h. After 20 h at room temperature (23–25 °C), doxifluridine and vehicle control were applied to the plate containing PA14 pathogenic bacteria, and about 120 synchronized D1 worms were added and cultured at 25 °C. The number of deaths was scored every 2–8 hours until all worms died. This experiment was repeated twice.

For hydrogen peroxide stress assay, 20 D1 worms were treated with drugs for 3 days and then added to a plate containing 10 mM $H_2O_2$ (Sigma, Cat#BCCF3861). The number of deaths was counted after 30 minutes treatment. two replicate experiments were performed.

## Intestinal atrophy measurement

The intestine of more than 16 worms was observed and imaged under the DIC of a microscope. For intestinal atrophy assay, we measured relative intestinal width behind the pharynx, which was calculated by intestinal width subtracting the luminal width and dividing by the body width as previously described [32]. This experiment was repeated twice.

## Muscle mitochondrial analysis

*C. elegans* muscle mitochondrial morphology is measured as previously described [31]. PD4251 worms were imaged at the microscope under a 100x objective. The mitochondria were divided into three types: the tubular mitochondria, the intermediate mitochondria (partial loss of tubular morphology) and the fragmented mitochondria (completely broken without tubular morphology). The number for each type of mitochondria in muscle cells were counted. The composition of mitochondria types in each experimental group was calculated and compared by GraphPad Prism 8.0, using Chi-square test to determine statistical significance. This experiment was repeated twice.

## Cell viability analysis

Cell viability was observed using the Cell Counting Kit-8 (CCK-8) Kit (Yeasen Biotechnology (Shanghai), Cat#40203ES76). LO2 cells ($5 \times 10^3$ cells/well, Procell Life Science&Technology) were cultured in 37 °C incubator overnight until the cells are fully adherent to the wall. Then, various doses of doxifluridine were added to each well. After 24 hours incubation, fresh complete medium containing Cell Counting Kit-8 (CCK-8) solution was added to the plates for 3 hours incubation. Absorbance was measured at 450 nm using a microplate absorbance reader. This experiment was repeated three times.

## Bacterial strains and culture conditions

Bacterial strains (*E. coli* OP50, BL21, MG1655, BW25113, X1666, HT115) were grown in LB without antibiotics. *E. coli* single deletion mutants from the Keio collection were grown in LB containing 100 µg/mL kanamycin. Δ*udp*Δ*upp*, Δ*gltD*Δ*gltB* and Δ*adiA*Δ*speA* double mutant

were cultured at LB appropriate supplemented with 100 μg/mL apramycin. Kanamycin and apramycin were not added into NGM plates when bacteria were used to feeding worms to avoid possible detrimental impact caused by antibiotic exposure.

## Bacterial mutant strains construction

Double mutant strains were obtained by knocking out single gene in *E. coli* BW25113 mutant strains. Strains with double mutations were constructed as the protocols described [55]. We used apramycin resistance cassette flanked by FRT (FLP recognition target) sites. Briefly, amplified PCR fragments >500 bp upstream and downstream of the predicted gene from BW25113 genomic DNA and sequenced nested fragments (upstream -FRT with apramycin resistance-downstream) and then directly introduced to high efficiency electroporation-competent cells which were constituted by single mutants carrying pKD46 plasmid. Apramycin-sensitive deletion mutants were confirmed by DNA sequence.

For overexpressing genes in bacteria, plasmids (pET28A-*yjjG*, *upp*, *udp*, *adiY* and *lrp*) were confirmed by sequence and transferred into the respective mutant (strains for *E. coli* OP50, *Δupp*, *Δudp*, and *Δlrp*, respectively) and used for worms to test lifespan. pETDuet-1 designed for co-expression of two target genes were used in double mutant strains *ΔudpΔupp* rescue experiment.

All the primers were listed in S9 Table.

## Production of bacterial powder, heat-killed OP50, UV-killed OP50

Bacterial powder was generated as a published protocol described [33]. Briefly, 2 L overnight bacterial culture was centrifuged for 20 minutes at 8000 g, 4 °C. Remove the supernatant except for the bacterial pellet. The bacterial pellets were then transferred to fresh tubes, washed three times in sterile water, frozen with liquid nitrogen and -80 °C for 6 hours, and placed in 80 °C oven overnight. Finally, we disrupted cells with sterile mortar-pestle in aseptic condition. Bacterial powder was dissolved in sterile water to achieve a concentration of 50 mg/mL.

Heat-killed OP50: Culture *E. coli* OP50 from a single clone overnight and dilute 1,000-fold in 30 ml LB, then shaking at 220 rpm for 16 hours. The bacteria were then heat-killed at 80 °C for 120 min. We finally concentrated the cultured OP50–1/10 volume and add 100 μL onto NGM plates.

UV-killed OP50: The culture plate seeded with OP50 bacteria (100 μL) was irradiated under UV light for 16 hours.

**Quantifying bacterial abundance on NGM plates.** 100 μL of *E. coli* OP50 was inoculated onto a 3 cm NGM plate and incubated at room temperature for 48 hours. After 48 hours, 400 μM doxifluridine (final concentration) was added to the bacterial lawn, and the plate was incubated at room temperature for an additional 20 hours. The bacterial lawn was then transferred to an EP tube using 2 mL of sterile water. The bacterial concentration was measured using a NanoDrop One/OneC at 600 nm (OD600) by reading 1 mL of the bacterial suspension. The relative bacterial concentration was calculated as: Relative measurement value = mean OD600 $\times$ 8 $\times$ 10$^5$ cells/μL [56]. Measurements were independently carried out three times.

**Measuring bacteria growth in liquid.** Pick a single colony of OP50 bacteria and culture it overnight in 10 mL LB medium at 37°C, 240 rpm/min. The next day, add 10 μL of bacterial culture to 10 mL of fresh LB medium with final concentrations of 400 μM doxifluridine, 5 μM carbenicillin, or a combination of both (doxifluridine 400 μM and carbenicillin 5 μM), and culture at 37°C, 240 rpm/min. For the control group, add the same volume of water.

Measure OD600 absorbance every hour until bacterial growth reaches a plateau, then stop measurements.

## Colony forming unit assays

Fresh *E. coli* OP50 (100 µL) was inoculated onto a 3 cm NGM plate. After 48 hours, treat the experimental groups with one of the following treatments (final concentrations): 400 µM doxifluridine, 5 µM carbenicillin, or a combination of 400 µM doxifluridine and 5 µM carbenicillin, applied directly to the bacterial surface of the plate. The control and positive control were supplemented with the same volume (100 µL) of water and 75% alcohol. Incubated the plates at room temperature for approximately 24 hours. After treatment, resuspend the bacteria in 2 mL of sterile water, measure the OD600 absorbance, and standardize it to OD600 = 0.1 using sterile water. Samples of the resuspended bacteria (standardized to OD600 = 0.1) were serially diluted (1:10,000), and 10 µL of each sample was streaked onto a 10 cm LB agar plate, which was then incubated at 37°C for 16 hours. The number of colonies formed was counted the next day.

## Bacterial genomic sequencing

Different *E. coli* strains were incubating in 15 mL LB overnight. Samples were centrifuged at 12,000 g for 4 minutes at 4 °C. The supernatant was removed and pellets were transferred to 1.5 mL microcentrifuge tube. The pellet was quickly frozen with liquid nitrogen and stored at -80 °C. 2 independent biological replicates were included per strain. The quality of isolated genomic DNA was verified by using these three methods in combination: (1) DNA degradation and contamination were monitored on 1% agarose gels; (2) DNA concentration was measured by Qubit DNA Assay Kit in Qubit 3.0 Flurometer (Invitrogen, USA).

## Library preparation

A total amount of 0.2 µg DNA per sample was used as input material for the DNA library preparations. Sequencing library was generated using NEB Next Ultra DNA Library Prep Kit for Illumina (NEB, USA) following manufacturer's recommendations and index codes were added to each sample. Briefly, genomic DNA sample was fragmented by sonication to a size of 350 bp. Then DNA fragments were endpolished. A-tailed, and ligated with the full-length adapter for Illumina sequencing, followed by further PCR amplification. After PCR products were purified by AMPure XP system (Beckman Coulter, Beverly, USA), DNA concentration was measured by Qubit3.0 Flurometer (Invitrogen, USA), libraries were analyzed for size distribution by NGS3K/Caliper and quantified by real-time PCR (3 nM). The original fluorescence image files obtained from Illumina platform are transformed to short reads (Raw data) by base calling and these short reads are recorded in FASTQ format, which contains sequence information and corresponding sequencing quality information.

## Bacterial proteomics

NGM plates were seeded with 300 µL of overnight bacterial culture and supplemented with vehicle or doxifluridine. 5–6 independent biological replicates were included per condition. Bacteria lawns were left to grow at 20 °C for 2 days and collected from plates using sterile water. Samples were centrifuged at 12,000 g for 4 minutes at 4 °C. The supernatant was removed and pellets were transferred to 1.5 mL microcentrifuge tube. 200 µL of chilled protein lysis buffer (Beyotime, P0013B) supplemented with Protease Inhibitor Cocktail (ThermoFisher, Cat#A32963) was added to the pellet. Samples were kept on ice from this point onward. Pellets were lysed via sonication for 2 x 10 s and proteins were separated from the

cellular debris by centrifuging at 12,000 g for 12 minutes at 4 °C. Supernatant containing the extracted protein was transferred to clean 1.5 mL microcentrifuge tube. The protein concentration in each sample was determined by Bradford assay (Bio-Rad 5000205). The extracted proteins (50 μg) from each sample were reduced with 10 mM TCEP at 56 °C for 1 hours and alkylated with 20 mM iodoacetamide at room temperature for 30 minutes in darkness. Protein samples were precipitated by the chloroform/methanol/water method and digested with trypsin (trypsin: protein = 1 μg: 50 μg) at 37 °C for 16 hours.

A TMT-based strategy was used for proteomic analyses of Bacteria. Bacteria were labeled with the TMT-10plex reagents (Thermo Scientific). Restore the MTT-10-Plex (0.8 mg) isotope labeling kit to room temperature. After centrifugation at high speed, 41 μL anhydrous Acetonitrile was added into each TMT labeling reagent (0.8 mg) and vortexed until fully dissolved. Take 10ug of each sample and add 8ul TMT-labeled reagent. Oscillate at room temperature for 1 hours. Each tube sample was added with 1.7 μL of 5% hydroxylamine.

## HPLC fractionation

To increase the depth of protein identification, TMT-labeled peptides were fractioned by using HPLC (SHIMADZU LC-2030 plus) on a 25 cm reversed-phase column (4 μm, 4.6 × 250 mm, Poroshell, Agilent) under basic pH conditions (pH = 10). The parameters were set as follows: the flow rate was 1 mL/min, the column temperature was 40 °C, and ultraviolet detection was performed at 214 nm. Gradient elution was performed on a mixture of buffer A (98% $H_2O$ and 2% ACN, 10 mM ammonium formate, pH = 10) and buffer B (90% ACN, 10% $H_2O$, 10 mM ammonium formate, pH = 10). The 120 min LC gradient was set as follows, 3%-35% B in 90 minutes, 35%-60% B in 15 minutes, 60%-100% B in 10 minutes, and 100%-3% B in 5 minutes. The 120 components are combined into 20 and then concentrated and dried for desalination.

## Mass spectrometry analysis

The desalted peptides were loaded onto a 75 μm (inner diameter) × 2.5 cm (length) trap column (Spursil C18 5 μm particle size, DIKMA) and analyzed on a 75 μm (inner diameter) × 25 cm (length) analytical column (Reprosil-Pur C18-AQ 1.9 μm particle size, Dr. Maisch). LC-MS/MS analysis was carried out using a Nano EASY-NLC 1000 nanoflow LC instrument coupled with a Q Exactive Plus Quadrupole-Orbitrap mass spectrometer (Thermo Fisher Scientific). Peptides were analyzed using a 65 min gradient of 6% to 95% Buffer B (0.1% FA in 95% ACN) at a flow rate of 330 nL/min. Data-dependent acquisition (DDA) was performed in positive ion mode. Full MS scans (m/z 350–1600) were acquired with a resolution of 60,000. The automatic gain control (AGC) value was set to 3e6, and the maximum injection time (MIT) was 20 ms. For MS/MS analysis, the top 20 most intense precursor ions were selected at an isolation window of 0.6 m/z and then fragmented with a normalized collision energy of 30%. The AGC value of MS/MS was set to 1e5, and the MIT was 64 ms. Precursor ions with charge states of z = 1 or 8 or an unassigned charge state were excluded. The dynamic exclusion duration was 50 s.

## MS database searching

Raw files for proteomics were searched using MaxQuant (version 1.6). Proteomic data of bacteria were searched against the *E. coli* BL21 protein sequence database (4806 protein sequences). The precursor peptide mass tolerance was 10 ppm, and the fragment ion mass tolerance was 0.02 Da. The minimum amino acid length was set to 6. Carbamidomethylation of cysteine was assigned as a fixed modification. Oxidation of methionine and protein

N-terminal acetylation were assigned as variable modifications. Two missed trypsin cleavages were allowed. Peptides with a false discovery rate (FDR) < 1% were kept for further data processing.

KEGG pathway was acquired using online DAVID enrichment analysis service. Transcription factor (TF) enrichment analysis was estimated using TF-gene association data from RegulonDB.

## Bacterial metabolomics

300 μL of overnight bacterial culture were plated on NGM plates and treated with vehicle or doxifluridine. 5–6 independent biological replicates were included per group. Bacteria lawns were treated at 20 °C for 2 days and collected from plates using sterile water. Samples were stored at -80 °C until metabolite extraction. 800 μL of cold methanol/acetonitrile (1:1, v/v) was added to remove the protein and extract the metabolites. The mixture was collected into a new centrifuge tube, and centrifuged at 14000 g for 5 minutes at 4 °C to collect the supernatant. The supernatant was dried in a vacuum centrifuge. For LC-MS analysis, the samples were re-dissolved in 100 μL acetonitrile/water (1:1, v/v) solvent.

For untargeted metabolomics of polar metabolites, extracts were analyzed using a quadrupole time-of-flight mass spectrometer (Sciex TripleTOF 6600) coupled to hydrophilic interaction chromatography via electrospray ionization in Shanghai Applied Protein Technology Co., Ltd. LC separation was on a ACQUIY UPLC BEH Amide column (2.1 mm × 100 mm, 1.7 μm particle size (waters, Ireland) using a gradient of solvent A (25 mM ammonium acetate and 25 mM ammonium hydroxide in water) solvent B (acetonitrile). The column temperature was 25 °C. The mass spectrometer was operated in both negative ion and positive ionizations mode.

For peak picking, the following parameters were used: centWave m/z = 25 ppm, peakwidth = c (10, 60), prefilter = c (10, 100). For peak grouping, bw = 5, mzwid = 0.025, minfrac = 0.5 were used. Compound identification of metabolites by MS/MS spectra with an in-house database established with available authentic standards. The variable importance in the projection (VIP) value of each variable in the OPLS-DA model was calculated to indicate its contribution to the classification. Significance was determined using an unpaired Student's t test. VIP value >1 and p<0.05 was considered as statistically significant.

## Statistics

For all graphs, the error bars indicate the mean ± s.e.m. To compare data from different groups, unpaired two-tailed Student's $t$ test, Two-way ANOVA with Sidak's multiple comparisons test, Two-way ANOVA with Dunnett's multiple comparisons test, One-way ANOVA with Dunnett's multiple comparisons test, Chi-square test or Kaplan-Meier method followed by the log-rank test were used as indicated in the Fig legends. All statistical analyses were performed using GraphPad Prism 8.0 software. Differences were considered significant if P < 0.05. Statistical significance is indicated by asterisks; *P < 0.05; **P < 0.01; ***P < 0.001; ****P < 0.0001.

## Supporting information

**S1 Fig. Pharmaceutical compounds were identified in the age-associated AS defects rescuing screen.** (A) Schematic of the ret-1 splicing reporter. (B) Left, representative fluorescent images showing the *ret-1* splicing reporter at Day 1 (D1) and Day 7 (D7). Right, quantification of GFP/mCherry ratio in (B). Each worm was measured by randomly selecting 8 regions in whole intestine. 6 worms were measured per group (Methods). n=2 independent experiments.

(C) Increased GFP/mCherry ratio in worms with 10 candidate drugs treatment compared to vehicle treatment. n=2 independent experiments. (D) Fluorescent images of worms with lanolin, doxifluridine, 7213–0276 treatment. n=2 independent experiments. (E-G) Lifespan curves of N2 worms treated with lanolin (E), doxifluridine (F), 7213–0276 (G). n=2 independent experiments. (H) Mean lifespan extension percent of two experiment in worms treated with lanolin, doxifluridine, 7213–0276. (I) Fluorescent images of splicing reporter worms at Day 7 (D7) treated with different concentrations of FUDR. n=2 independent experiments. (J) GFP/mCherry fluorescent intensity ratio in I. Around 10 worms were measured per group (Methods). n=2 independent experiments. (K) Inhibition ratio of human hepatic cell line LO2 growth by different concentrations of doxifluridine and FUDR. Summary of three independent experiments, with each point representing one replicate in S1K Fig. n=3 independent experiments. (L) Normalized ratio of GFP/mCherry in metformin, rapamycin and doxifluridine-treated worms. n=2 independent experiments. The lifespan data illustrated in the figure corresponds to one repeat experiment. Error bars, SEM. *$P < 0.05$, **$P < 0.01$, ***$P < 0.001$, ****$P < 0.0001$. B, C, J, K, unpaired two-tailed Student's t test. E-G, Log-rank (Mantel-Cox). L, one-way ANOVA. B, D, I, scale bars, 100 μm.
(TIF)

**S2 Fig. The effect of doxifluridine on lifespan extension requires bacteria metabolism.** (A) Lifespan curves of worms treated with or without doxifluridine 40 μM in UV-killed bacteria. n=2 independent experiments. (B) Lifespan curves of worms in doxifluridine pretreated with OP50. n=2 independent experiments. (C) The growth curves of OP50 in doxifluridine (400 μM), carbenicillin (5 μM), and a blank control. Combination of three independent experiments. n=3 independent experiments. (D) Relative colony-forming units (CFUs) of OP50 bacteria after treatment with 0 μM, 5 μM, and 10 μM carbenicillin, with or without the addition of doxifluridine. n=2 independent experiments. (E) Mean lifespan extension of worms treated with doxifluridine (400 μM) in different laboratory bacteria strains. *E. coli* K-12 strains: BW25113, HT115, MG1655, X1666; *E. coli* B strains: OP50, BL21. (F) Lifespan curves of worms treated with or without doxifluridine 400 μM in BW25113 and BW25113 △*tdk*. n=2 independent experiments. (G) Fluorescent images of splicing reporter worms with or without doxfluridine 400 μM treatment in wildtype, Δ*upp*Δ*udp*, Δ*gltB*Δ*gltD* BW25113 bacteria. n=2 independent experiments. (H) GFP/mCherry fluorescent intensity ratio in G (9–10 worms were measured per group). n=2 independent experiments. (I) Relative lifespan extension percent in worms treated with doxifluridine 40 μM in OP50 adding with nucleosides (adenine, thymidine, cytidine, guanine and uridine) respectively, normalized to OP50 (CTRL) group. (J and K) Lifespan curves in worms treated with or without doxifluridine 40 μM in OP50 adding with uridine (J), thymidine (K) respectively. n=2 independent experiments. Lifespan curves of OP50 treated with and without doxifluridine 40 μM were shown twice in (J) and (K) for comparison. The lifespan data illustrated in the figure corresponds to one repeat experiment. Error bars, SEM. A, B, F, J, K, Log-rank (Mantel-Cox) test. D, E, H, unpaired two-tailed Student's t test. G, scale bars, 100 μm.
(TIF)

**S3 Fig. Bacterial metabolic pathways affect doxifluridine efficacy on lifespan.** (A) Diagram displaying the RegulonDB transcription factors (TF) that regulate the metabolic pathway from proteomics data in E. coli OP50 treated with doxifluridine. (B) Relative lifespan extension of worms treated with doxifluridine in TF-mutant bacteria groups, normalized to BW25113 (WT) group. (C) Lifespan curves of worms treated with or without doxifluridine 40 μM in OP50 with or without glutamine supplementation. n=2 independent experiments. (D) PCA plot of *E. coli* OP50 metabolomics data showing effect of doxifluridine-treatment (400 μM) or

vehicle-treatment (0 µM) in *E. coli* OP50. (E) Lifespan curves of worms treated with methysticin (91.15 µM), deoxyinosine (436 µM), inosine (913.35 µM) or vehicle. n=2 independent experiments. (F) Fluorescent images of splicing reporter worms with or without doxfluridine 400 µM treatment in OP50, OP50 with 3mg/mL glutamine. n=2 independent experiments. (G) Relative GFP/mCherry fluorescent intensity ratio in F (9–10 worms were measured per group). n=2 independent experiments. (H) Fluorescent images of splicing reporter worms treat with linoleic acid (75 µM) and agmatine (25 mM). n=2 independent experiments. (I) Relative GFP/mCherry fluorescent intensity ratio in H (9–10 worms were measured per group). n=2 independent experiments. The lifespan data shown in the figure is one of the two replicates. C, E, Log-rank (Mantel-Cox) test. G, I, unpaired two-tailed Student's t test. F, H, scale bars, 100 µm

(TIF)

**S1 Table. Drug specific information.**
(XLSX)

**S2 Table. GO analysis of differential alternative splicing genes.**
(XLSX)

**S3 Table. Lifespan analysis.**
(XLSX)

**S4 Table. Bacteria unique high mutation gene list.**
(XLSX)

**S5 Table. Differential protein levels.**
(XLSX)

**S6 Table. Transcription factors.**
(XLSX)

**S7 Table. Differential metabolites.**
(XLSX)

**S8 Table. Bacteria and strains list.**
(XLSX)

**S9 Table. Primers.**
(XLSX)

**S1 Data. Source data.**
(XLSX)

## Acknowledgments

We thank Dr. William Mair at Harvard T. H. Chan School of Public Health providing strain KH2235. Na Jiang (Advanced Mass spectrometry Center in West China Hospital) work on the metabolite examination, Bin Chen for technical support. Several *C. elegans* strains used in this work were provided by the Caenorhabditis Genetics Center (CGC).

## Author contributions

**Conceptualization:** Yuan Wang, Yan Zhang.

**Data curation:** Rui Wei, Yuling Peng.

**Formal analysis:** Rui Wei, Yuling Peng, Yamei Luo, Huan Yang, Yaojia Liu.

**Funding acquisition:** Yuan Wang, Yan Zhang.

**Investigation:** Yuan Wang, Yan Zhang.

**Methodology:** Yuling Peng, Yamei Luo, Xinyuan Wang, Zhengzhong Pan, Ran Zhou, Huan Yang, Zongyao Huang.

**Project administration:** Rui Wei, Yuan Wang.

**Resources:** Xinyuan Wang, Zhengzhong Pan, Ran Zhou, Zongyao Huang, Lunzhi Dai.

**Software:** Zhengzhong Pan, Ran Zhou, Zongyao Huang, Lunzhi Dai.

**Supervision:** Yan Zhang.

**Validation:** Yamei Luo.

**Writing – original draft:** Yuan Wang, Yan Zhang.

**Writing – review & editing:** Rui Wei, Yuling Peng, Yamei Luo, Yuan Wang, Yan Zhang.

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
