## [Decision Letter · Decision Letter 0]

9 Dec 2024

PGENETICS-D-24-01277

Doxifluridine promotes host longevity through bacterial metabolism

PLOS Genetics

Dear Dr. Zhang,

Thank you for submitting your manuscript to PLOS Genetics. After careful consideration, we feel that it has merit but does not fully meet PLOS Genetics's publication criteria as it currently stands. Therefore, we invite you to submit a revised version of the manuscript that addresses the points raised during the review process.

Please submit your revised manuscript within 60 days Feb 07 2025 11:59PM. If you will need more time than this to complete your revisions, please reply to this message or contact the journal office at plosgenetics@plos.org. Please include the following items when submitting your revised manuscript:

We look forward to receiving your revised manuscript.

Kind regards,

Daniel A. Starr

Academic Editor

PLOS Genetics

Fengwei Yu

Section Editor

PLOS Genetics

Aimée Dudley

Editor-in-Chief

PLOS Genetics

Anne Goriely

Editor-in-Chief

PLOS Genetics

**Additional Editor Comments:**

Thank you for this manuscript. You will need to address all the major comments of all three reviewers, which will require additional experimentation and increases in sample size, before resubmitting.

**Journal Requirements:**

At this stage, the following Authors/Authors require contributions: Rui Wei, Yuling Peng, Yamei Luo, Xinyuan Wang, Zhengzhong Pan, Huan Yang, Zongyao Huang, Yaojia Liu, Lunzhi Dai, and Yuan Wang. Please ensure that the full contributions of each author are acknowledged in the "Add/Edit/Remove Authors" section of our submission form.

The list of CRediT author contributions may be found here: https://journals.plos.org/plosgenetics/s/authorship#loc-author-contributions

2) We ask that a manuscript source file is provided at Revision. Please upload your manuscript file as a .doc, .docx, .rtf or .tex. If you are providing a .tex file, please upload it under the item type LaTeX Source File and leave your .pdf version as the item type Manuscript.

https://journals.plos.org/plosgenetics/s/submission-guidelines#loc-parts-of-a-submission

4) We do not publish any copyright or trademark symbols that usually accompany proprietary names, eg ©,  ®, or TM  (e.g. next to drug or reagent names). Therefore please remove all instances of trademark/copyright symbols throughout the text, including:

- ® on pages: 15, and 16

- TM on pages: 14, and 15.

5) Please upload all main figures as separate Figure files in .tif or .eps format. For more information about how to convert and format your figure files please see our guidelines: 

6) We have noticed that you have uploaded Supporting Information files, but you have not included a list of legends. Please add a full list of legends for your Supporting Information files after the references list.

7) We notice that your supplementary Figures are included in the manuscript file. Please remove them and upload them with the file type 'Supporting Information'. Please ensure that each Supporting Information file has a legend listed in the manuscript after the references list.

8) Some material included in your submission may be copyrighted. According to PLOSu2019s copyright policy, authors who use figures or other material (e.g., graphics, clipart, maps) from another author or copyright holder must demonstrate or obtain permission to publish this material under the Creative Commons Attribution 4.0 International (CC BY 4.0) License used by PLOS journals. Please closely review the details of PLOSu2019s copyright requirements here: PLOS Licenses and Copyright. If you need to request permissions from a copyright holder, you may use PLOS's Copyright Content Permission form.

Potential Copyright Issues:

i) Figures 1A, and 5. Please confirm whether you drew the images / clip-art within the figure panels by hand. If you did not draw the images, please provide (a) a link to the source of the images or icons and their license / terms of use; or (b) written permission from the copyright holder to publish the images or icons under our CC BY 4.0 license. Alternatively, you may replace the images with open source alternatives. See these open source resources you may use to replace images / clip-art:

9)Thank you for stating that " Genome sequencing raw data of different E. coli strains have been deposited in the Sequence Read Archive under accession number PRJNA1070821. The mass spectrometry proteomics data have been deposited in the ProteomeXchange Consortium via the iProX partner repository30,31 with dataset identifier PXD049160." Please note that, though access restrictions are acceptable now, your entire minimal dataset will need to be made freely accessible if your manuscript is accepted for publication. This policy applies to all data except where public deposition would breach compliance with the protocol approved by your research ethics board. If you are unable to adhere to our open data policy, please kindly revise your statement to explain your reasoning and we will seek the editor's input on an exemption.

10) Please amend your detailed Financial Disclosure statement. This is published with the article. It must therefore be completed in full sentences and contain the exact wording you wish to be published.

11) Your current Financial Disclosure states, "National Clinical Research Center for Geriatrics, West China Hospital, Sichuan University (Z2021JC006), National Natural Science Foundation of China(32471214).".

However, your funding information on the submission form indicates only one funder. Please ensure that the funders and grant numbers match between the Financial Disclosure field and the Funding Information tab in your submission form. Note that the funders must be provided in the same order in both places as well.

Please indicate by return email the full and correct funding information for your study and confirm the order in which funding contributions should appear. Please be sure to indicate whether the funders played any role in the study design, data collection and analysis, decision to publish, or preparation of the manuscript.

**Reviewers' comments:**

Reviewer's Responses to Questions

Reviewer #1: This manuscript describes an interesting phenomenon in which the nucleotide analog doxifluridine increases C. elegans lifespan. The authors show that live bacterial cultures with specific wild-type nucleotide metabolism enzymes are necessary for this extension. Although many questions remain about the specific links between drug administration and organismal phenotypes, the manuscript is for the most part well-supported and convincing.

Major comments:

1. It is important to explain more about the reporter system used for screening, in both the results, figure, and figure legend. What gene is being targeted? Which exon(s) of that gene? Is there something special about this alternative splicing that is coupled to lifespan? Or are all alternative splicing events likely to have similar patterns with aging and similar responses to drug administration?

2. The authors claim in the abstract and intro that they “demonstrate its mechanism of action” by which doxifluridine increases C. elegans lifespan. I find this claim to be an overreach, as by no means do we leave with a clear molecular mechanistic pathway leading from doxifluridine to organismal lifespan. We have a few observations (live bacteria are required, certain bacterial genes are required), but much work would remain to support the claim to “demonstration of the mechanism of action.”

3. Greater clarity and impact would be achieved if a clearer link could be made between the initial screened phenotype (splicing reporter) and the phenotype followed for the rest of the paper (lifespan). Are there any conditions in which these two phenotypes can be decoupled? Is one phenotype ‘upstream’ of the other? i.e. does mis-splicing lead to aging, does aging lead to mis-splicing, or are both phenotypes a consequence of some other upstream initiator? This would go a long way toward strengthening the mechanism of action claim above. Practical experiments that could partially address this question would be (a) live bacteria are required for the lifespan phenotype of doxifluridine; what about the splicing reporter phenotype? (b) same question about the ribonucleotide metabolism mutants; (c) same question but about glutamine (and other metabolite) supplementation. Additional experiments would be required for a full mechanistic explanation, but these would be a good (and easy) first step.

Minor comments:

4. The use of the word “consistently” is inconsistently and sometimes incorrectly used. Sometimes it is used correctly, in the sense of “invariably.” But other times it is used incorrectly, in the sense of “this is consistent with our hypothesis.” If the latter meaning is intended, a different phrase should be used. Otherwise the implication is that “these data are reproducible or invariable across trials,” without showing any such supporting data. Lines 104, 178, 222 are examples of mis-use.

5. Line 62 claims that aging is driven by disrupted transcriptional and protein homeostasis. I’d recommend a more nuanced and well-supported claim, such as “…disrupted transcriptional and protein homeostasis contributes to aging…”

Reviewer #2: In the article entitle ‘Doxifluridine promotes host longevity through bacterial metabolism’, the authors conducted a screen covering thousands of chemicals aiming at identifying potential aging-combating medication. The novelty of using the efficient nematode model and its medical implications are impressive. However, a few points should be fully addressed to be scientifically convincing enough, as listed below.

Major points:

The association between alternative splicing defects and aging does not automatically justify using florescence as a phenotypic readout of longevity, as evidenced by the low overlapping ratio (3/10, Figure 1A, Table S1) presented by this article. Either a more appropriate screening method should be adopted, or the interpretations thus the purpose/significance of this screen should be modified accordingly.

The lifespan of C. elegans observed in this study is substantially shorter than most reported in the principle. The N2 worms feeding on live OP50 at 20C typically exhibit an average lifespan of 17-19 days (e.g. PMID：17081160；PMID：23415229). Plus, the data show that the population starts to decease at Day 5-6 (e.g. Figure 1D), even well before complete cessation of reproduction, which is so unlikely that it makes all the longevity information across the article questionable. Possible reasons for this aberrant lifespan include: 1) involuntary artificial selection on the N2 population; 2) rough manipulation of the experimental animals that results in physical injuries; 3) contamination of the cultivating medium by harmful environmental germs. The authors didn’t provide the detailed information on animal maintenance or the sterility techniques used during screens and validation, making estimation on the likelihood of above reasons improbable at this point. Anyway, this lifespan issue should be solved to ensure that any conclusions obtained indeed relevant to the aging processes.

It is well acknowledged that killing or inactivation of bacteria is sufficient to lengthen lifespan of C. elegans. Before concluding that doxifluridine is a longevity-promoting molecule, the authors should rule out the possibility that this compound directly limits survival of symbiotic E. coli— in other words, it is simply an antibiotic. Monitoring influences of high-concentration doxifluridine on the bacterial growth curves would be helpful.

Minor points:

* The authors should provide information on criteria or methods of judging solubility of the drugs. It is even not clear whether doxifluridine is water or DMSO controlled.

* It should be clarified how the progeny of experimental animals were excluded from analyses during the screen.

* The rationals behind bacterial strain selection (e.g. OP50 vs. BW25113) at each point should be explained.

Reviewer #3: This interesting and well-written paper reports the identification of the compound doxifluridine as a potential avenue for rescuing age-associated alternative splicing defects and extending lifespan. The authors use a C. elegans screening platform to screen a large number of compounds and characterize their top hit doxifluridine. They show that doxifluridine has a strong efficacy at preventing age-associated AS decline in C. elegans, increasing resistance to multiple types of stress, improving age-related intestinal phenotypes, and extending lifespan. They further find that the lifespan extension by doxifluridine requires bacterial metabolism and identify several specific bacterial pathways including ribonucleotide metabolism, nitrogen metabolism, and agamatine synthesis that are required. I have some concerns regarding the number of replicates for certain experiments and certain information that is missing from the Methods, but overall the authors’ conclusions are supported by the data presented. The study is well conceived and likely to be of interest to the aging research community, especially because doxifluridine has already been entered into phase II and III clinical trials in elderly populations.

Major concerns:

My most substantive concern is that a few of the experiments appear to have been performed only a single time and were not replicated. In particular for Figure 1C, the figure legend states that “6-8 worms were measured per group” and references the Methods, but I could not find any information in Methods as to whether this experiment had been performed a single time with 6-8 worms, or had been repeated with independent biological replicates. This also applies to the muscle mitochondrial analysis (Figure 1H-I) and the intestinal width analysis (Figure 1J), which also do not list in the Methods or in the figure legend if it was replicated. In addition for Figure 1J the Methods states that 30 worms were imaged but the figure legend states that n=16. If these experiments have been replicated, that information should be provided in the Methods and in the figure legends. If they have not been, then the experiments should be repeated (I would recommend at least twice more, given the very low n) to ensure that the results are reproducible.

My other major concern is a lack of detail in the Methods sections for the drug screen and for the microscopy and image analysis. For the drug screen, it is not clear which method was used to synchronize the nematodes. The reporter analysis was also not conducted until D7 of adulthood after plating the worms as D1 adults, and it is not clear how the authors differentiated between the originally plated worms and progeny after 7 days or if sterility was induced somehow. In the microscopy section, the only details provided for the GFP/mCherry quantification are “Images were analyzed in ImageJ” (line 325). Additional detail is needed here including how ROIs for quantification were selected, whether any background correction was applied, and whether mean pixel intensity or integrated density was used for calculating fluorescence.

Minor concerns:

The paper would be strengthened significantly if the authors could show a connection between the rescue of age-associated AS defects by doxifluridine and the effects of doxifluiridine on lifespan, for instance by showing that the downstream bacterial pathways they identified as required for the doxifluridine-mediated LS extension were also required for the effect on AS. As it stands, it is possible that the lifespan extension by doxifluridine and the rescue of age-associated decline in AS are via completely separate mechanisms. However, since the authors are not making the claim in this paper that it is the drug’s rescue of age-associated AS defects that leads to the lifespan extension, this is merely a suggestion. They may wish to save this topic to be explored in future studies.

For Figure 1E and 1F I was able to confirm from Table S2 that these experiments were repeated twice – however, this information should also be available in the relevant Methods sections and/or in the figure legends, which it currently is not.

Although the Methods and/or figure legends state that the lifespan experiments were repeated twice, it is unclear for all of the lifespan figures whether the data shown in the figure is one of the two replicates, or both combined. This should be clarified in the figure legends.

In the Figure 2 legend it says “Error bars, SEM” but there are no error bars shown in this figure.

In Figure 2, the authors are using the same controls in both Figure 2A and 2B, but in Table S2 it says that it is 2B and 3A that are the same.

In Figure 3, it is hard to tell which pairs of lifespans are statistically different from each other as p values are not provided in the figure or in the legend. This is also true in the text where they do give the p values (lines 173-175), but it is not entirely clear from the text which comparisons are being made to produce those values.

**Have all data underlying the figures and results presented in the manuscript been provided?**

Reviewer #1: Yes

Reviewer #2: Yes

Reviewer #3: Yes

PLOS authors have the option to publish the peer review history of their article (what does this mean? ). If published, this will include your full peer review and any attached files.

**Do you want your identity to be public for this peer review?** For information about this choice, including consent withdrawal, please see our Privacy Policy .

Reviewer #1: No

Reviewer #2: No

Reviewer #3: No

**Figure resubmission:**
---

## [Decision Letter · Decision Letter 1]

9 Mar 2025

Dear Dr Zhang,

We are pleased to inform you that your manuscript entitled "Doxifluridine promotes host longevity through bacterial metabolism" has been editorially accepted for publication in PLOS Genetics. Congratulations!

Yours sincerely,

Daniel A. Starr

Academic Editor

PLOS Genetics

Fengwei Yu

Section Editor

PLOS Genetics

Aimée Dudley

Editor-in-Chief

PLOS Genetics

Anne Goriely

Editor-in-Chief

PLOS Genetics

Comments from the reviewers (if applicable):

Reviewer's Responses to Questions

**Comments to the Authors:**

Reviewer #1: The authors have done a great job of responding to my review. In particular, I think the new data in figure 2, S2, and S3 are really cool and strengthen the idea that the bacterial metabolism is tightly linked with both aging and splicing regulation, even if the exact details remain an open question.

Reviewer #2: The authors addressed most of my questions satisfactorily.

**Have all data underlying the figures and results presented in the manuscript been provided?**

Reviewer #1: Yes

Reviewer #2: Yes

PLOS authors have the option to publish the peer review history of their article (what does this mean? ). If published, this will include your full peer review and any attached files.

**Do you want your identity to be public for this peer review?** For information about this choice, including consent withdrawal, please see our Privacy Policy .

Reviewer #1: No

Reviewer #2: No

**Data Deposition**

http://datadryad.org/submit?journalID=pgenetics&manu=PGENETICS-D-24-01277R1

**Press Queries**

---

## [Editor Report · Acceptance letter]

PGENETICS-D-24-01277R1

Doxifluridine promotes host longevity through bacterial metabolism

Dear Dr Zhang,

We are pleased to inform you that your manuscript entitled "Doxifluridine promotes host longevity through bacterial metabolism" has been formally accepted for publication in PLOS Genetics! Your manuscript is now with our production department and you will be notified of the publication date in due course.

With kind regards,

Anita Estes

PLOS Genetics

On behalf of:
